# Words or Characters? Fine-grained Gating for Reading Comprehension

**Zhilin Yang, Bhuwan Dhingra, Ye Yuan, Junjie Hu, William W. Cohen, Ruslan Salakhutdinov**
School of Computer Science
Carnegie Mellon University
{zhiliny,wcohen,rsalakhu}@cs.cmu.edu

## Abstract

Previous work combines word-level and character-level representations using concatenation or scalar weighting, which is suboptimal for high-level tasks like reading comprehension. We present a fine-grained gating mechanism to dynamically combine word-level and character-level representations based on properties of the words. We also extend the idea of fine-grained gating to modeling the interaction between questions and paragraphs for reading comprehension. Experiments show that our approach can improve the performance on reading comprehension tasks, achieving new state-of-the-art results on the Children's Book Test and Who Did What datasets. To demonstrate the generality of our gating mechanism, we also show improved results on a social media tag prediction task.[1]

## 1 Introduction

Finding semantically meaningful representations of the words (also called *tokens*) in a document is necessary for strong performance in Natural Language Processing tasks. In neural networks, tokens are mainly represented in two ways, either using word-level representations or character-level representations. Word-level representations are obtained from a lookup table, where each unique token is represented as a vector. Character-level representations are usually obtained by applying recurrent neural networks (RNNs) or convolutional neural networks (CNNs) on the character sequence of the token, and their hidden states are combined to form the representation. Word-level representations are good at memorizing the semantics of the tokens while character-level representations are more suitable for modeling sub-word morphologies (Ling et al., 2015; Yang et al., 2016a). For example, considering "cat" and "cats", word-level representations can only learn the similarities between the two tokens by training on a large amount of training data, while character-level representations, by design, can easily capture the similarities. Character-level representations are also used to alleviate the difficulties of modeling out-of-vocabulary (OOV) tokens (Luong & Manning, 2016).

Hybrid word-character models have been proposed to leverage the advantages of both word-level and character-level representations. The most commonly used method is to concatenate these two representations (Yang et al., 2016a). However, concatenating word-level and character-level representations is technically problematic. For frequent tokens, the word-level representations are usually accurately estimated during the training process, and thus introducing character-level representations can potentially bias the entire representations. For infrequent tokens, the estimation of word-level representations have high variance, which will have negative effects when combined with the character-level representations. To address this issue, recently Miyamoto & Cho (2016) introduced a scalar gate conditioned on the word-level representations to control the ratio of the two representations. However, for the task of reading comprehension, preliminary experiments showed that this method was not able to improve the performance over concatenation. There are two possible reasons. First, word-level representations might not contain sufficient information to support the decisions of selecting between the two representations. Second, using a scalar gate means applying the same ratio for each of the dimensions, which can be suboptimal.

In this work, we present a fine-grained gating mechanism to combine the word-level and character-level representations. We compute a vector gate as a linear projection of the token features followed

---

[1]Code is available at https://github.com/kimiyoung/fg-gating

by a sigmoid activation. We then multiplicatively apply the gate to the character-level and word-level representations. Each dimension of the gate controls how much information is flowed from the word-level and character-level representations respectively. We use named entity tags, part-of-speech tags, document frequencies, and word-level representations as the features for token properties which determine the gate. More generally, our fine-grained gating mechanism can be used to model multiple levels of structure in language, including words, characters, phrases, sentences and paragraphs. In this work we focus on studying the effects on word-character gating.

To better tackle the problem of reading comprehension, we also extend the idea of fine-grained gating for modeling the interaction between documents and queries. Previous work has shown the importance of modeling interactions between document and query tokens by introducing various attention architectures for the task (Hermann et al., 2015; Kadlec et al., 2016). Most of these use an inner product between the two representations to compute the relative importance of document tokens. The Gated-Attention Reader (Dhingra et al., 2016a) showed improved performance by replacing the inner-product with an element-wise product to allow for better matching at the semantic level. However, they use aggregated representations of the query which may lead to loss of information. In this work we use a fine-grained gating mechanism for each token in the paragraph and each token in the query. The fine-grained gating mechanism applies an element-wise multiplication of the two representations.

We show improved performance on reading comprehension datasets, including Children's Book Test (CBT), Who Did What, and SQuAD. On CBT, our approach achieves new state-of-the-art results without using an ensemble. Our model also improves over state-of-the-art results on the Who Did What dataset. To demonstrate the generality of our method, we apply our word-character fine-grained gating mechanism to a social media tag prediction task and show improved performance over previous methods.

Our contributions are two-fold. First, we present a fine-grained word-character gating mechanism and show improved performance on a variety of tasks including reading comprehension. Second, to better tackle the reading comprehension tasks, we extend our fine-grained gating approach to modeling the interaction between documents and queries.

## 2 RELATED WORK

Hybrid word-character models have been proposed to take advantages of both word-level and character-level representations. Ling et al. (2015) introduce a compositional character to word (C2W) model based on bidirectional LSTMs. Kim et al. (2016) describe a model that employs a convolutional neural network (CNN) and a highway network over characters for language modeling. Miyamoto & Cho (2016) use a gate to adaptively find the optimal mixture of the character-level and word-level inputs. Yang et al. (2016a) employ deep gated recurrent units on both character and word levels to encode morphology and context information. Concurrent to our work, Rei et al. (2016) employed a similar gating idea to combine word-level and character-level representations, but their focus is on low-level sequence tagging tasks and the gate is not conditioned on linguistic features.

The gating mechanism is widely used in sequence modeling. Long short-term memory (LSTM) networks (Hochreiter & Schmidhuber, 1997) are designed to deal with vanishing gradients through the gating mechanism. Similar to LSTM, Gated Recurrent Unit (GRU) was proposed by Cho et al. (2014), which also uses gating units to modulate the flow of information. The gating mechanism can also be viewed as a form of attention mechanism (Bahdanau et al., 2015; Yang et al., 2016b) over two inputs.

Similar to the idea of gating, multiplicative integration has also been shown to provide a benefit in various settings. Yang et al. (2014) find that multiplicative operations are superior to additive operations in modeling relations. Wu et al. (2016) propose to use Hadamard product to replace sum operation in recurrent networks, which gives a significant performance boost over existing RNN models. Dhingra et al. (2016a) use a multiplicative gating mechanism to achieve state-of-the-art results on question answering benchmarks.

Reading comprehension is a challenging task for machines. A variety of models have been proposed to extract answers from given text (Hill et al., 2016; Kadlec et al., 2016; Trischler et al., 2016; Chen

et al., 2016; Sordoni et al., 2016; Cui et al., 2016). Yu et al. (2016) propose a dynamic chunk reader to extract and rank a set of answer candidates from a given document to answer questions. Wang & Jiang (2016) introduce an end-to-end neural architecture which incorporates match-LSTM and pointer networks (Vinyals et al., 2015).

## 3 FINE-GRAINED GATING

In this section, we will describe our fine-grained gating approach in the context of reading comprehension. We first introduce the settings of reading comprehension tasks and a general neural network architecture. We will then describe our word-character gating and document-query gating approaches respectively.

### 3.1 READING COMPREHENSION SETTING

The reading comprehension task involves a document $P = (p_1, p_2, \cdots, p_M)$ and a query $Q = (q_1, q_2, \cdots, q_N)$, where $M$ and $N$ are the lengths of the document and the query respectively. Each token $p_i$ is denoted as $(\mathbf{w}_i, \mathbf{C}_i)$, where $\mathbf{w}_i$ is a one-hot encoding of the token in the vocabulary and $\mathbf{C}_i$ is a matrix with each row representing a one-hot encoding of a character. Each token in the query $q_j$ is similarly defined. We use $i$ as a subscript for documents and $j$ for queries. The output of the problem is an answer $a$, which can either be an index or a span of indices in the document.

Now we describe a general architecture used in this work, which is a generalization of the gated attention reader (Dhingra et al., 2016a). For each token in the document and the query, we compute a vector representation using a function $f$. More specifically, for each token $p_i$ in the document, we have $\mathbf{h}_i^0 = f(\mathbf{w}_i, \mathbf{C}_i)$. The same function $f$ is also applied to the tokens in the query. Let $\mathbf{H}_p^0$ and $\mathbf{H}_q$ denote the vector representations computed by $f$ for tokens in documents and queries respectively. In Section 3.2, we will discuss the "word-character" fine-grained gating used to define the function $f$.

Suppose that we have a network of $K$ layers. At the $k$-th layer, we apply RNNs on $\mathbf{H}_p^{k-1}$ and $\mathbf{H}_q$ to obtain hidden states $\mathbf{P}^k$ and $\mathbf{Q}^k$, where $\mathbf{P}^k$ is a $M \times d$ matrix and $\mathbf{Q}^k$ is a $N \times d$ matrix with $d$ being the number of hidden units in the RNNs. Then we use a function $r$ to compute a new representation for the document $\mathbf{H}_p^k = r(\mathbf{P}^k, \mathbf{Q}^k)$. In Section 3.3, we will introduce the "document-query" fine-grained gating used to define the function $r$.

After going through $K$ layers, we predict the answer index $a$ using a softmax layer over hidden states $\mathbf{H}_p^k$. For datasets where the answer is a span of text, we use two softmax layers for the start and end indices respectively.

### 3.2 WORD-CHARACTER FINE-GRAINED GATING

Given a one-hot encoding $\mathbf{w}_i$ and a character sequence $\mathbf{C}_i$, we now describe how to compute the vector representation $\mathbf{h}_i = f(\mathbf{w}_i, \mathbf{C}_i)$ for the token. In the rest of the section, we will drop the subscript $i$ for notation simplicity.

We first apply an RNN on $\mathbf{C}$ and take the hidden state in the last time step $\mathbf{c}$ as the character-level representation (Yang et al., 2016a). Let $\mathbf{E}$ denote the token embedding lookup table. We perform a matrix-vector multiplication $\mathbf{Ew}$ to obtain a word-level representation. We assume $\mathbf{c}$ and $\mathbf{Ew}$ have the same length $d_e$ in this work.

Previous methods defined $f$ using the word-level representation $\mathbf{Ew}$ (Collobert et al., 2011), the character-level representation $\mathbf{c}$ (Ling et al., 2015), or the concatenation $[\mathbf{Ew}; \mathbf{c}]$ (Yang et al., 2016a). Unlike these methods, we propose to use a gate to dynamically choose between the word-level and character-level representations based on the properties of the token. Let $\mathbf{v}$ denote a feature vector that encodes these properties. In this work, we use the concatenation of named entity tags, part-of-speech tags, binned document frequency vectors, and the word-level representations to form the feature vector $\mathbf{v}$. Let $d_v$ denote the length of $\mathbf{v}$.

The gate is computed as follows:
$$\mathbf{g} = \sigma(\mathbf{W}_g \mathbf{v} + \mathbf{b}_g)$$

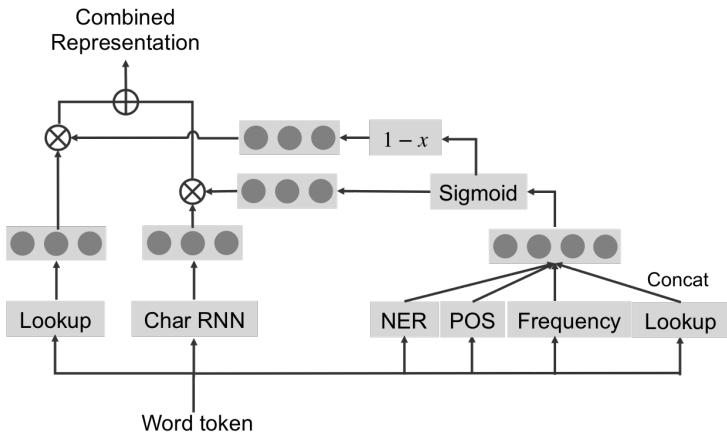

Figure 1: Word-character fine-grained gating. The two lookup tables are shared. "NER", "POS", "frequency" refer to named entity tags, part-of-speech tags, document frequency features.

where $\mathbf{W}_g$ and $\mathbf{b}_g$ are the model parameters with shapes $d_e \times d_v$ and $d_e$, and $\sigma$ denotes an element-wise sigmoid function.

The final representation is computed using a fine-grained gating mechanism,

$$\mathbf{h} = f(\mathbf{c}, \mathbf{w}) = \mathbf{g} \odot \mathbf{c} + (1 - \mathbf{g}) \odot (\mathbf{Ew})$$

where $\odot$ denotes element-wise product between two vectors.

An illustration of our fine-grained gating mechanism is shown in Figure 1. Intuitively speaking, when the gate $\mathbf{g}$ has high values, more information flows from the character-level representation to the final representation; when the gate $\mathbf{g}$ has low values, the final representation is dominated by the word-level representation.

Though Miyamoto & Cho (2016) also use a gate to choose between word-level and character-level representations, our method is different in two ways. First, we use a more fine-grained gating mechanism, i.e., vector gates rather than scalar gates. Second, we condition the gate on features that better reflect the properties of the token. For example, for noun phrases and entities, we would expect the gate to bias towards character-level representations because noun phrases and entities are usually less common and display richer morphological structure. Experiments show that these changes are key to the performance improvements for reading comprehension tasks.

Our approach can be further generalized to a setting of multi-level networks so that we can combine multiple levels of representations using fine-grained gating mechanisms, which we leave for future work.

### 3.3 DOCUMENT-QUERY FINE-GRAINED GATING

Given the hidden states $\mathbf{P}^k$ and $\mathbf{Q}^k$, we now describe how to compute a representation $\mathbf{H}^k$ that encodes the interactions between the document and the query. In this section, we drop the superscript $k$ (the layer number) for notation simplicity. Let $\mathbf{p}_i$ denote the $i$-th row of $\mathbf{P}$ and $\mathbf{q}_j$ denote the $j$-row of $\mathbf{Q}$. Let $d_h$ denote the lengths of $\mathbf{p}_i$ and $\mathbf{q}_j$.

Attention-over-attention (AoA) (Cui et al., 2016) defines a dot product between each pair of tokens in the document and the query, i.e., $\mathbf{p}_i^T \mathbf{q}_j$, followed by row-wise and column-wise softmax non-linearities. AoA imposes pair-wise interactions between the document and the query, but using a dot product is potentially not expressive enough and hard to generalize to multi-layer networks. The gated attention (GA) reader (Dhingra et al., 2016a) defines an element-wise product as $\mathbf{p}_i \odot \mathbf{g}_i$ where $\mathbf{g}_i$ is a gate computed by attention mechanism on the token $p_i$ and the entire query. The intuition for the gate $\mathbf{g}_i$ is to attend to important information in the document. However, there is no direct pair-wise interaction between each token pair.

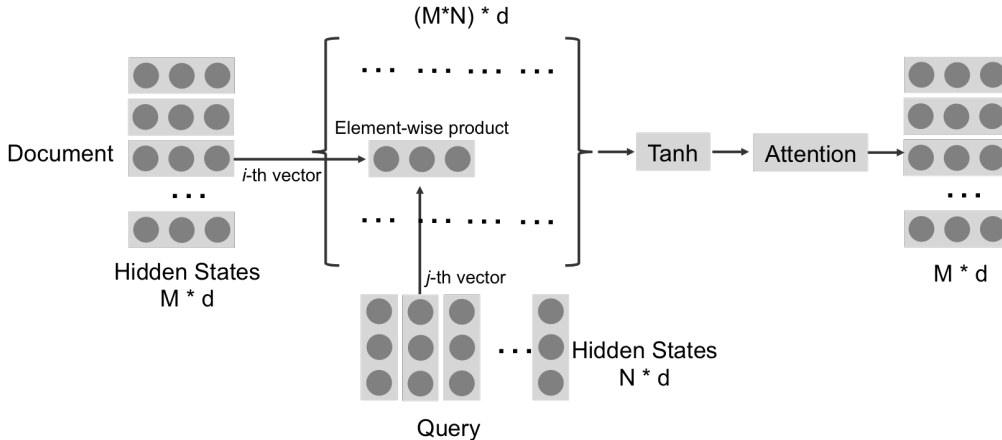

Figure 2: Paragraph-question fine-grained gating.

We present a fine-grained gating method that combines the advantages of the above methods (i.e., both pairwise and element-wise). We compute the pairwise element-wise product between the hidden states in the document and the query, as shown in Figure 2. More specifically, for $p_i$ and $q_j$, we have

$$\mathbf{I}_{ij} = \tanh(\mathbf{p}_i \odot \mathbf{q}_j)$$

where $\mathbf{q}_j$ can be viewed as a gate to filter the information in $\mathbf{p}_i$. We then use an attention mechanism over $\mathbf{I}_{ij}$ to output hidden states $\mathbf{h}_i$ as follows

$$\mathbf{h}_i = \sum_j \text{softmax}(\mathbf{u}_h^T \mathbf{I}_{ij} + \mathbf{w}_i^T \mathbf{w}_j b_{h1} + b_{h2}) \mathbf{I}_{ij}$$

where $\mathbf{u}_h$ is a $d_v$-dimensional model parameter, $b_{h1}$ and $b_{h2}$ are scalar model parameters, $\mathbf{w}_i$ and $\mathbf{w}_j$ are one-hot encodings for $p_i$ and $q_j$ respectively. We additionally use one-hot encodings in the attention mechanism to reinforce the matching between the same tokens since such information is not fully preserved in $\mathbf{I}_{ij}$ when $k$ is large. The softmax nonlinearity is applied over all $j$'s. The final hidden states $\mathbf{H}$ are formed by concatenating the $\mathbf{h}_i$'s for each token $p_i$.

## 4 EXPERIMENTS

We first present experimental results on the Twitter dataset where we can rule out the effects of different choices of network architectures, to demonstrate the effectiveness of our word-character fine-grained gating approach. Later we show experiments on more challenging datasets on reading comprehension to further show that our approach can be used to improve the performance on high-level NLP tasks as well.

### 4.1 EVALUATING WORD-CHARACTER GATING ON TWITTER

We evaluate the effectiveness of our word-character fine-grained gating mechanism on a social media tag prediction task. We use the Twitter dataset and follow the experimental settings in Dhingra et al. (2016b). We also use the same network architecture upon the token representations, which is an LSTM layer followed by a softmax classification layer (Dhingra et al., 2016b). The Twitter dataset consists of English tweets with at least one hashtag from Twitter. Hashtags and HTML tags have been removed from the body of the tweet, and user names and URLs are replaced with special tokens. The dataset contains 2 million tweets for training, 10K for validation and 50K for testing, with a total of 2,039 distinct hashtags. The task is to predict the hashtags of each tweet.

We compare several different methods as follows. **Word char concat** uses the concatenation of word-level and character-level representations as in Yang et al. (2016a); **word char feat concat** concatenates the word-level and character-level representations along with the features described in

Table 1: Performance on the Twitter dataset. "word" and "char" means using word-level and character-level representations respectively.

| Model | Precision@1 | Recall@10 | Mean Rank |
|---|---|---|---|
| word (Dhingra et al., 2016b) | 0.241 | 0.428 | 133 |
| char (Dhingra et al., 2016b) | 0.284 | 0.485 | 104 |
| word char concat | 0.2961 | 0.4959 | 105.8 |
| word char feat concat | 0.2951 | 0.4974 | 106.2 |
| scalar gate | 0.2974 | 0.4982 | 104.2 |
| fine-grained gate | **0.3069** | **0.5119** | **101.5** |

Table 2: Performance on the CBT dataset. The "GA word char concat" results are extracted from Dhingra et al. (2016a). Our results on fine-grained gating are based on a single model. "CN" and "NE" are two widely used question categories. "dev" means development set, and "test" means test set.

| Model | CN dev | CN test | NE dev | NE test |
|---|---|---|---|---|
| GA word char concat | 0.731 | 0.696 | 0.768 | 0.725 |
| GA word char feat concat | 0.7250 | 0.6928 | 0.7815 | 0.7256 |
| GA scalar gate | 0.7240 | 0.6908 | 0.7810 | 0.7260 |
| GA fine-grained gate | 0.7425 | 0.7084 | 0.7890 | 0.7464 |
| FG fine-grained gate | **0.7530** | **0.7204** | **0.7910** | **0.7496** |
| | | | | |
| Sordoni et al. (2016) | 0.721 | 0.692 | 0.752 | 0.686 |
| Trischler et al. (2016) | 0.715 | 0.674 | 0.753 | 0.697 |
| Cui et al. (2016) | 0.722 | 0.694 | 0.778 | 0.720 |
| Munkhdalai & Yu (2016) | 0.743 | 0.719 | 0.782 | 0.732 |
| | | | | |
| Kadlec et al. (2016) ensemble | 0.711 | 0.689 | 0.762 | 0.710 |
| Sordoni et al. (2016) ensemble | 0.741 | 0.710 | 0.769 | 0.720 |
| Trischler et al. (2016) ensemble | 0.736 | 0.706 | 0.766 | 0.718 |

Section 3.2; **scalar gate** uses a scalar gate similar to Miyamoto & Cho (2016) but is conditioned on the features; **fine-grained gate** is our method described in Section 3.2. We include word char feat concat for a fair comparison because our fine-grained gating approach also uses the token features.

The results are shown in Table 1. We report three evaluation metrics including precision@1, recall@10, and mean rank. Our method outperforms character-level models used in Dhingra et al. (2016b) by 2.29%, 2.69%, and 2.5 points in terms of precision, recall and mean rank respectively. We can observe that scalar gating approach (Miyamoto & Cho, 2016) can only marginally improve over the baseline methods, while fine-grained gating methods can substantially improve model performance. Note that directly concatenating the token features with the character-level and word-level representations does not boost the performance, but using the token features to compute a gate (as done in fine-grained gating) leads to better results. This indicates that the benefit of fine-grained gating mainly comes from better modeling rather than using additional features.

## 4.2 PERFORMANCE ON READING COMPREHENSION

After investigating the effectiveness of the word-character fine-grained gating mechanism on the Twitter dataset, we now move on to a more challenging task, reading comprehension. In this section, we experiment with two datasets, the Children's Book Test dataset (Hill et al., 2016) and the SQuAD dataset (Rajpurkar et al., 2016).

### 4.2.1 CLOZE-STYLE QUESTIONS

We evaluate our model on cloze-style question answering benchmarks.

Table 3: Performance on the Who Did What dataset. "dev" means development set, and "test" means test set. "WDW-R" is the relaxed version of WDW.

| Model | WDW dev | WDW test | WDW-R dev | WDW-R test |
|---|---|---|---|---|
| Kadlec et al. (2016) | – | 0.570 | – | 0.590 |
| Chen et al. (2016) | – | 0.640 | – | 0.650 |
| Munkhdalai & Yu (2016) | 0.665 | 0.662 | 0.670 | 0.667 |
| Dhingra et al. (2016a) | 0.716 | 0.712 | 0.726 | **0.726** |
| this paper | **0.723** | **0.717** | **0.731** | **0.726** |

Table 4: Performance on the SQuAD dev set. Test set results are included in the brackets.

| Model | F1 | Exact Match |
|---|---|---|
| GA word | 0.6695 | 0.5492 |
| GA word char concat | 0.6857 | 0.5639 |
| GA word char feat concat | 0.6904 | 0.5711 |
| GA scalar gate | 0.6850 | 0.5620 |
| GA fine-grained gate | 0.6983 | 0.5804 |
| FG fine-grained gate | 0.7125 | 0.5995 |
| FG fine-grained gate + ensemble | 0.7341 (0.733) | 0.6238 (0.625) |
| Yu et al. (2016) | 0.712 (0.710) | 0.625 (0.625) |
| Wang & Jiang (2016) | 0.700 (0.703) | 0.591 (0.595) |

The Children's Book Test (CBT) dataset is built from children's books. The whole dataset has 669,343 questions for training, 8,000 for validation and 10,000 for testing. We closely follow the setting in Dhingra et al. (2016a) and incrementally add different components to see the changes in performance. For the fine-grained gating approach, we use the same hyper-parameters as in Dhingra et al. (2016a) except that we use a character-level GRU with 100 units to be of the same size as the word lookup table. The word embeddings are updated during training.

In addition to different ways of combining word-level and character-level representations, we also compare two different ways of integrating documents and queries: **GA** refers to the gated attention reader (Dhingra et al., 2016a) and **FG** refers to our fine-grained gating described in Section 3.3.

The results are reported in Table 2. We report the results on common noun (CN) questions and named entity (NE) questions, which are two widely used question categories in CBT. Our fine-grained gating approach achieves new state-of-the-art performance on both settings and outperforms the current state-of-the-art results by up to 1.76% without using ensembles. Our method outperforms the baseline GA reader by up to 2.4%, which indicates the effectiveness of the fine-grained gating mechanism. Consistent with the results on the Twitter dataset, using word-character fine-grained gating can substantially improve the performance over concatenation or scalar gating. Furthermore, we can see that document-query fine-grained gating also contributes significantly to the final results.

We also apply our fine-grained gating model to the Who Did What (WDW) dataset (Onishi et al., 2016). As shown in Table 3, our model achieves state-of-the-art results compared to strong baselines. We fix the word embeddings during training.

### 4.2.2 SQUAD

The Stanford Question Answering Dataset (SQuAD) is a reading comprehension dataset collected recently (Rajpurkar et al., 2016). It contains 23,215 paragraphs come from 536 Wikipedia articles. Unlike other reading comprehension datasets such as CBT, the answers are a span of text rather than a single word. The dataset is partitioned into a training set (80%, 87,636 question-answer pairs), a development set (10%, 10,600 question-answer pairs) and a test set which is not released.

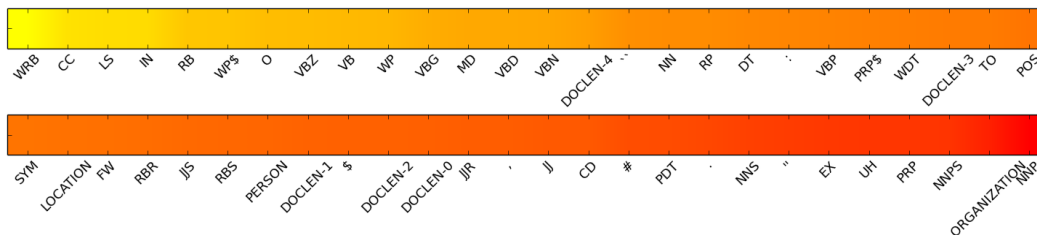

Figure 3: Visualization of the weight matrix $\mathbf{W}_g$. Weights for each features are averaged. Red means high and yellow means low. High weight values favor character-level representations, and low weight values favor word-level representations. "Organization", "'Person", "Location", and "O" are named entity tags; "DOCLEN-n" are document frequency features (larger $n$ means higher frequency, $n$ from 0 to 4); others are POS tags.

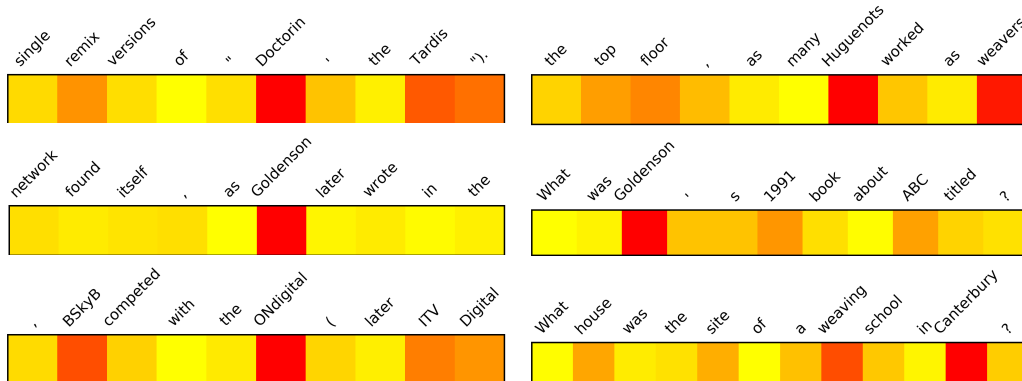

Figure 4: Visualization of gate values in the text. Red means high and yellow means low. High gate values favor character-level representations, and low gate values favor word-level representations.

We report our results in Table 4. "Exact match" computes the ratio of questions that are answered correctly by strict string comparison, and the F1 score is computed on the token level. We can observe that both word-character fine-grained gating and document-query fine-grained gating can substantially improve the performance, leading to state-of-the-art results among published papers. Note that at the time of submission, the best score on the leaderboard is 0.716 in exact match and 0.804 in F1 without published papers. A gap exists because our architecture described in Section 3.1 does not specifically model the answer span structure that is unique to SQuAD. In this work, we focus on this general architecture to study the effectiveness of fine-grained gating mechanisms.

## 4.3 VISUALIZATION AND ANALYSIS

We visualize the model parameter $\mathbf{W}_g$ as described in Section 3.2. For each feature, we average the corresponding weight vector in $\mathbf{W}_g$. The results are described in Figure 3. We can see that named entities like "Organization" and noun phrases (with tags "NNP" or "NNPS") tend to use character-level representations, which is consistent with human intuition because those tokens are usually infrequent or display rich morphologies. Also, DOCLEN-4, WH-adverb ("WRB"), and conjunction ("IN" and "CC") tokens tend to use word-level representations because they appear frequently.

We also sample random span of text from the SQuAD dataset, and visualize the average gate values in Figure 4. The results are consistent with our observations in Figure 3. Rare tokens, noun phrases, and named entities tend to use character-level representations, while others tend to use word-level representations. To further justify this argument, we also list the tokens with highest and lowest gate values in Table 5.

Table 5: Word tokens with highest and lowest gate values. High gate values favor character-level representations, and low gate values favor word-level representations.

| Gate values | Word tokens |
| --- | --- |
| Lowest | or but But These these However however among Among that when When although Although because Because until many Many than though Though this This Since since date where Where have That and And Such such number so which by By how before Before with With between Between even Even if |
| Highest | Sweetgum Untersee Jianlong Floresta Chlorella Obersee PhT Doctorin Jumonville WFTS WTSP Boven Pharm Nederrijn Otrar Rhin Magicicada WBKB Tanzler KMBC WPLG Mainau Merwede RMJM Kleitman Scheur Bodensee Kromme Horenbout Vorderrhein Chlamydomonas Scantlebury Qingshui Funchess |

## 5 CONCLUSIONS

We present a fine-grained gating mechanism that dynamically combines word-level and character-level representations based on word properties. Experiments on the Twitter tag prediction dataset show that fine-grained gating substantially outperforms scalar gating and concatenation. Our method also improves the performance on reading comprehension and achieves new state-of-the-art results on CBT and WDW. In our future work, we plan to to apply the fine-grained gating mechanism for combining other levels of representations, such as phrases and sentences. It will also be intriguing to integrate NER and POS networks and learn the token representation in an end-to-end manner.

ACKNOWLEDGMENTS

This work was funded by NVIDIA, the Office of Naval Research Scene Understanding grant N000141310721, the NSF grant IIS1250956, and Google Research.

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
