# Peer review of "Words or Characters? Fine-grained Gating for Reading Comprehension"

_ICLR 2017 — accepted_

[Official Review · AnonReviewer1 · rating 7 · confidence 3 · 15 Dec 2016]
**No Title**

SUMMARY.

The paper proposes a gating mechanism to combine word embeddings with character-level word representations.
The gating mechanism uses features associated to a word to decided which word representation is the most useful.
The fine-grain gating is applied as part of systems which seek to solve the task of cloze-style reading comprehension question answering, and Twitter hashtag prediction.
For the question answering task, a fine-grained reformulation of gated attention for combining document words and questions is proposed.
In both tasks the fine-grain gating helps to get better accuracy, outperforming state-of-the-art methods on the CBT dataset and performing on-par with state-of-the-art approach on the SQuAD dataset.


----------

OVERALL JUDGMENT

This paper proposes a clever fine-grained extension of a scalar gate for combining word representation.
It is clear and well written. It covers all the necessary prior work and compares the proposed method with previous similar models.

I liked the ablation study that shows quite clearly the impact of individual contributions.
And I also liked the fact that some (shallow) linguistic prior knowledge e.g., pos tags ner tags, frequency etc. has been used in a clever way. 
It would be interesting to see if syntactic features can be helpful.

[Official Review · AnonReviewer2 · rating 7 · confidence 4 · 17 Dec 2016]
clarity 3 · substance 4

This paper proposes a new gating mechanism to combine word and character representations. The proposed model sets a new state-of-the-art on the CBT dataset; the new gating mechanism also improves over scalar gates without linguistic features on SQuAD and a twitter classification task. 

Intuitively, the vector-based gate working better than the scalar gate is unsurprising, as it is more similar to LSTM and GRU gates. The real contribution of the paper for me is that using features such as POS tags and NER help learn better gates. The visualization in Figure 3 and examples in Table 4 effectively confirm the utility of these features, very nice! 

In sum, while the proposed gate is nothing technically groundbreaking, the paper presents a very focused contribution that I think will be useful to the NLP community. Thus, I hope it is accepted.

[Official Review · AnonReviewer3 · rating 6 · confidence 4 · 17 Dec 2016]
**No Title**
soundness 5 · clarity 4 · impact 3

I think the problem here is well motivated, the approach is insightful and intuitive, and the results are convincing of the approach (although lacking in variety of applications). I like the fact that the authors use POS and NER in terms of an intermediate signal for the decision. Also they compare against a sufficient range of baselines to show the effectiveness of the proposed model.

I am also convinced by the authors' answers to my question, I think there is sufficient evidence provided in the results to show the effectiveness of the inductive bias introduced by the fine-grained gating model.

[Public Comment · Kris Cao · 07 Feb 2017]
**Prior work on this area**

Sorry to pop up late, but I've been looking over ICLR accepted papers, and I noticed this one. Something very similar, by combining word and character information at the feature level using sigmoid gating, has been done before, see

[Final Decision · Program Chairs · 06 Feb 2017]
**ICLR committee final decision**

The consensus amongst reviewers is that this paper's proposal for combining character level information with word-level information is sound, well presented, and well evaluated. The main negative sentiment was that the approach was perhaps a little increment, although the conceptual size of the increment was sufficient to warrant publication, and will be relevant to the interests of ICLR attendees working on the intersection of deep learning and NLP. After a cursory reading of the paper, I see no reason to disagree, and recommend acceptance.